# Targeted Delivery of Immunostimulatory CpG Oligodeoxynucleotides to Antigen-Presenting Cells in Draining Lymph Nodes by Stearic Acid Modification and Nanostructurization

**DOI:** 10.3390/ijms23031350

**Published:** 2022-01-25

**Authors:** Makoto Nagaoka, Wenqing Liao, Kosuke Kusamori, Makiya Nishikawa

**Affiliations:** Laboratory of Biopharmaceutics, Faculty of Pharmaceutical Sciences, Tokyo University of Science, 2641 Yamazaki, Noda 278-8510, Chiba, Japan; 3B20552@ed.tus.ac.jp (M.N.); wenqing.liao@outlook.com (W.L.); kusamori@rs.tus.ac.jp (K.K.)

**Keywords:** stearic acid, nanostructured DNA, Toll-like receptor 9, cytosine-phosphate-guanine oligodeoxynucleotide, antigen-presenting cell

## Abstract

Polypod-like structured nucleic acids (polypodnas), which are nanostructured DNAs, are useful for delivering cytosine-phosphate guanine oligodeoxynucleotides (CpG ODNs) to antigen-presenting cells (APCs) expressing Toll-like receptor 9 (TLR9) for immune stimulation. Lipid modification is another approach to deliver ODNs to lymph nodes, where TLR9-positive APCs are abundant, by binding to serum albumin. The combination of these two methods can be useful for delivering CpG ODNs to lymph nodes in vivo. In the present study, CpG1668, a phosphodiester-type CpG ODN, was modified with stearic acid (SA) to obtain SA-CpG1668. Tripodna, a polypodna with three pods, was selected as the nanostructured DNA. Tripodnas loaded with CpG1668 or SA-CpG1668 were obtained in high yields. SA-CpG1668/tripodna bound more efficiently to plasma proteins than CpG1668/tripodna and was more efficiently taken up by macrophage-like RAW264.7 cells than CpG1668/tripodna, whereas the levels of tumor necrosis factor-α released from the cells were comparable between the two. After subcutaneous injection into mice, SA-CpG1668/tripodna induced significantly higher interleukin (IL)-12 p40 production in the draining lymph nodes than SA-CpG1668 or CpG1668/tripodna, with reduced IL-6 levels in plasma. These results indicate that the combination of SA modification and nanostructurization is a useful approach for the targeted delivery of CpG ODNs to lymph nodes.

## 1. Introduction

The development of oligonucleotide therapeutics has recently accelerated, and the number of approved oligonucleotide therapeutics has been increasing. The classes of oligonucleotide therapeutics that have been approved include antisense oligonucleotide, small interfering RNAs, nucleic acid aptamers, and unmethylated cytosine-phosphate-guanine (CpG) oligodeoxynucleotides (ODNs). CpG ODN is a ligand for Toll-like receptor 9 (TLR9) [1,2,3], which is expressed on antigen-presenting cells (APCs), such as dendritic cells and macrophages. CpG ODNs stimulate innate immunity by being taken up by these cells and binding to TLR9 in endosomes [4,5,6].

The first CpG ODN approved for clinical use was CpG 1018, which is included as an adjuvant for a licensed vaccine for hepatitis B [7,8]. CpG 1018 is a DNase-resistant, chemically synthesized ODN with a phosphorothioate (PS) backbone. Despite the clinical development of PS-modified CpG ODNs, several challenges still remain, including the inefficient delivery of CpG ODNs to TLR9-positive cells, limited cellular uptake, and hepatic and renal toxicity [9].

One of the issues to be solved is the inefficient delivery of CpG ODNs to target cells. After administration, CpG ODNs need to be delivered to lymph nodes, where APCs present antigens to T lymphocytes [10]. Nanoparticles have been developed and used to deliver CpG ODNs to lymph nodes [11]; however, developing effective nanoparticles is a laborious process. These nanoparticles should at least have the following properties: a surface with minimal non-specific interaction with tissues, efficient encapsulation of CpG ONDs, and ligands for APC delivery. Therefore, an alternative to nanoparticles is expected to be developed as a safe and efficient delivery system for CpG ODNs to APCs in lymph nodes [12,13].

Ligand modification has attracted significant attention from the viewpoint of quality control and usability. Fatty acid modification of CpG ODNs has been reported to be useful for the delivery of CpG ODNs to APCs in draining lymph nodes [14,15,16]. After subcutaneous administration, diacyl lipid-modified CpG ODNs bound to serum albumin and were transported to lymph nodes. These results indicate that lipid modification can be useful for the delivery of ODNs to APCs.

Nanostructurization is another useful strategy for improving the uptake of ODNs by APCs. Previously, we demonstrated that polypod-like structured nucleic acids (polypodnas), consisting of three or more ODNs, were quite useful for the delivery of CpG ODNs to APCs. Compared to conventional single-stranded CpG ODNs, CpG ODNs loaded on polypodna (CpG ODN/polypodna) resulted in higher cellular uptake and subsequent proinflammatory cytokine production [17,18,19,20]. Macrophage scavenger receptor 1, or scavenger receptor class A type I, is involved in the uptake of polypodnas by bone marrow-derived dendritic cells and peritoneal macrophages [21,22]. In addition, this enhanced immunostimulation was observed not only with PS-modified CpG ODNs, but also with natural phosphodiester ODNs. Therefore, the nanostructurization of CpG ODNs could solve the problems associated with PS-modified CpG ODNs.

The results obtained thus far raise the possibility that highly efficient delivery of CpG ODNs to APCs can be achieved by the combination of lipid modification and nanostructurization. In the present study, we verified this possibility by developing stearic acid (SA)-modified CpG ODNs and tripodna, a polypodna consisting of three ODNs. CpG1668, a CpG ODN with a phosphodiester backbone, was selected and modified with SA to obtain SA-modified CpG1668 (SA-CpG1668). Then, tripodnas loaded with CpG1668 or SA-CpG1668 were prepared by annealing the ODNs. Then, their binding to plasma components and interaction with mouse macrophage-like RAW264.7 cells were evaluated. The immunostimulatory activities of CpG1668 and SA-CpG1668 were also evaluated after the subcutaneous injection into mice in the single-stranded form or after loading into the tripodna.

## 2. Results

### 2.1. Synthesis of SA-CpG1668

Figure 1 shows the synthesis of SA-CpG1668. SA-CpG1668 was synthesized in two steps. SA *N*-hydroxysuccinimide ester (SA-NHS) was obtained by reacting stearoyl chloride with *N*-hydroxysuccinimide, and SA-NHS was coupled to the amino group of 5′-NH_2_-CpG1668 to obtain SA-CpG1668. The yield of SA-CpG1668 determined by matrix-assisted laser desorption ionization–time of flight-mass spectrometry (MALDI–TOF-MS) was approximately 50%.

### 2.2. Binding to Bovine Serum Albumin (BSA)

CpG1668 or SA-CpG1668 was mixed with various concentrations of BSA and subjected to polyacrylamide gel electrophoresis (PAGE) (Figure 2A and Appendix A). In the case of CpG1668, the band of free CpG1668 was detected in all lanes where CpG1668 was mixed with 0–40 equivalents of BSA. On the other hand, the band of free SA-CpG1668 became faint depending on the concentration of BSA, and the band was hardly seen at 40 equivalents of BSA. The intensity of the bands was quantified and normalized with respect to CpG1668 or SA-CpG1668 in the absence of BSA (Figure 2B). This quantification showed that SA-CpG1668 bound more significantly to BSA than CpG1668.

### 2.3. Formation of SA-CpG1668/Tripodna

Tripodna was constructed by using Tri-1, Tri-2, and Tri-3, as previously reported (Table 1) [23]. Then CpG1668 or SA-CpG1668 was added to tripodna in a stoichiometric ratio of 3:1 to obtain CpG1668/tripodna and SA-CpG1668/tripodna, respectively. Figure 3A shows the schematic representation of the putative structures of CpG1668/tripodna and SA-CpG1668/tripodna. PAGE showed that all DNA samples had a single major band (Figure 3B). The addition of CpG1668 or SA-CpG1668 shifted the band of the tripodna upward, indicating the binding or loading of CpG1668 and SA-CpG1668 to tripodna. The migration distance of SA-CpG1668/tripodna was shorter than that of CpG1668/tripodna. These results indicate that SA-CpG1668/tripodna was successfully formed as CpG1668/tripodna.

### 2.4. Binding to Plasma Components

Mouse plasma was added at varying ratios to CpG1668/tripodna or SA-CpG1668/tripodna at a concentration of 0.1 pmol DNA/well (Figure 4A,B). PAGE showed that the band of free CpG1668/tripodna was detected even when CpG1668/tripodna was mixed with 80% plasma. In contrast, the band of free SA-CpG1668/tripodna was barely detected when SA-CpG1668/tripodna was mixed with 20% plasma, and SA-CpG1668/tripodna remained in the loading wells. There were smear broad bands in the lanes of DNA-free mouse plasma (Appendix A), indicating serum proteins were detected as smear bands in the gels. These results indicate that SA-CpG1668/tripodna, in contrast to CpG1668/tripodna, efficiently bound to plasma components. The stability of CpG1668/tripodna and SA-CpG1668/tripodna in 10% plasma was assessed by PAGE. The band of CpG1668/tripodna decreased with time. On the other hand, the band of SA-CpG1668/tripodna was detected in the loading wells, and the band intensity hardly changed at least for 4 h (Appendix A).

### 2.5. Thermal Stability under Simulated Extracellular and Endosomal Conditions

The thermal stability of CpG1668/tripodna and SA-CpG1668/tripodna was evaluated by measuring the melting temperature (Tm). Table 2 summarizes the Tm values of tripodnas and CpG1668 or SA-CpG1668 mixed with Tri-1, each of which formed a partial double-stranded structure. The Tm values of these samples at pH 7.4 ranged from 52 to 66 °C, while those at pH 5.5 ranged from 50 to 60 °C. When CpG1668 and SA-CpG1668 were compared, the Tm values of SA-CpG1668 were 4 (tripodna) or 9 °C (+tri-1) higher than those of CpG1668. These results indicate that SA modification increased the stability of CpG1668 to its complementary ODN.

### 2.6. Uptake by RAW264.7 Cells

It was reported that the cellular uptake of ODNs was increased by modification with lipids [24]. To compare the uptake of DNA samples by fluorescence intensity, Alexa Fluor568-labeled tri-1 was used for all DNA samples. CpG1668/Alexa Fluor568-labeled tri-1 and SA-CpG1668/Alexa Fluor568-labeled tri-1 were also prepared to estimate the uptake of single-stranded CpG1668 and SA-CpG1668. Figure 5 shows confocal microscopic images of RAW264.7 cells after addition of Alexa Fluor568-labeled DNA samples. A small fluorescence signal was observed in cells treated with CpG1668/Alexa Fluor568-labeled tri-1, whereas some fluorescence signals were detected after the addition of SA-CpG1668/Alexa Fluor568-labeled tri-1. Both CpG1668/Alexa Fluor568-labeled tripodna and SA-CpG1668/Alexa Fluor568-labeled tripodna were more efficiently taken up by the cells than CpG1668/Alexa Fluor568-labeled tri-1 or SA-CpG1668/Alexa Fluor568-labeled tri-1, respectively. The highest fluorescence signals were observed in cells treated with SA-CpG1668/Alexa Fluor568-labeled tripodna.

### 2.7. Production of the Tumor Necrosis Factor (TNF)-α after Addition to RAW264.7 Cells

Figure 6 shows the release of TNF-α from RAW264.7 cells at 8 h after the addition of DNA samples. SA-CpG1668 induced significantly higher TNF-α release from RAW264.7 cells than CpG1668. In both cases of CpG1668 and SA-CpG1668, the formation of tripodna, that is, CpG1668/tripodna and SA-CpG1668/tripodna, significantly increased TNF-α production in RAW264.7 cells. However, SA-CpG1668/tripodna was slightly less effective in inducing TNF-α release than CpG1668/tripodna.

### 2.8. IL-12p40 Production in Inguinal Lymph Nodes

SA-CpG1668 induced slightly greater IL-12p40 production in the inguinal lymph nodes than CpG1668 after subcutaneous injection into mice, but the difference was not statistically significant (Figure 7). In contrast, SA-CpG1668/tripodna induced a significant increase in IL-12p40 production among the samples, including CpG1668/tripodna.

### 2.9. IL-6 Production in Plasma

Excessive release of cytokines into the systemic circulation leads to cytokine storm syndrome. To evaluate the cytokine levels in the systemic circulation, the IL-6 concentration in plasma was measured after the subcutaneous injection of DNA samples (Figure 8). Plasma IL-6 levels increased with a peak at 3 h after injection of the DNA samples, and the levels were not statistically significant among the groups. However, the level of SA-CpG1668/tripodna was approximately 34% of that of CpG1668/tripodna.

### 2.10. Accumulation of PI-Labeled CpG1668/Tripodna and SA-CpG1668/Tripodna in Lymph Nodes

After administration of PI-labeled CpG1668/tripodna, no fluorescence signals were detected in any of the lymph nodes (Figure 9). On the other hand, the signals were detected in the left inguinal lymph node after at 1, 3, and 6 h after administration of PI-labeled SA-CpG1668/tripodna, with the highest at 6 h. The delivery to the inguinal lymph node can be explained by the fact that the administration site was in the caudal region close to the inguinal lymph node. The injection sites may possibly be slightly shifted to the left (not in the direct center), which could explain the biased delivery of SA-CpG1668/tripodna to the lymph nodes on the left side.

## 3. Discussion

One of the important issues in the clinical application of oligonucleotide therapeutics is improving the efficiency of the delivery of therapeutics to their targets. We have previously shown that polypodna, which are nanostructured DNAs with fixed structural properties, can be an efficient delivery vehicle for CpG ODNs to cultured APCs, such as dendritic cells and macrophages. Target APCs reside in lymph nodes, and attempts have been made to deliver CpG ODNs to lymph nodes by lipid modification. Nanostructurization and lipid modification are useful strategies for the delivery of CpG ODNs to target APCs, but few attempts have been made to combine these two for the delivery of CpG ODNs. Therefore, in the present study, we examined whether the combination of these two approaches was useful for the delivery of CpG ODNs to lymph nodes after subcutaneous injection into mice.

Previous studies on ODN modified with cholesterol or fatty acids have shown that lipid-modified ODNs form micelles in aqueous media [25,26,27]. PAGE showed that SA-CpG1668 hardly formed micelles in PBS. This could be explained by the fact that SA is less lipophilic than cholesterol or other highly lipophilic compounds, and that one SA molecule was coupled to CpG1668. SA-modified G-quadruplexes prepared with SA-modified ODNs formed micelles, probably because SA was arranged at high density in SA-modified G-quadruplexes. SA would be dispersedly located in SA-CpG1668/tripodna because the pods of tripodna extrude away from one another. Therefore, the conjugated SA molecule can interact with serum albumin or other plasma components after in vivo administration.

PS modification is the most frequently used chemical modification for oligonucleotide therapeutics. PS-modified ODNs have a high binding affinity for various proteins, including serum albumin and membrane proteins [28]. Our previous study, which used HEK Blue-hTLR9 cells, demonstrated that natural phosphodiester ODNs only interacted with cells transduced with macrophage scavenger receptor 1, whereas PS-modified ODNs were efficiently internalized by the cells [21]. These results indicate that PS-modified ODN binds to membrane proteins in a relatively non-specific manner.

SA modification significantly increased the binding affinity of CpG1668 to BSA, indicating that the conjugated SA contributed to binding. It has been reported that SA binds to the fatty-acid-binding sites of serum albumin [29,30,31]. The higher the amount of BSA added, the more SA-CpG1668 bound (Figure 2), suggesting that the binding affinity of SA-CpG1668 to BSA was moderate. Under in vivo conditions, SA-CpG1668 binds to serum albumin because the concentration of serum albumin is as high as 27 mg/mL [32]. We found that SA-CpG1668/tripodna bound to plasma proteins in plasma after intravenous injection in mice (Appendix A). Taken together, a CpG ODN derivative with binding affinity to serum albumin was developed.

SA-CpG1668/tripodna was successfully prepared with a yield similar to that of CpG1668 (Figure 3), suggesting that the conjugated SA hardly disturbed the loading to the tripodna. The size of the tripodna used in the present study would be about 7–10 nm, according to the measurement of polypodna, using small-angle X-ray scattering [33]. All the DNA samples were negatively charged (Appendix A), and CpG1668/tripodna and SA-CpG1668/tripodna were more negatively charged than single- or double-stranded ODNs [34]. Then the SA molecules of SA-CpG1668/tripodna seem to protrude from the nanostructure, which would be suitable for interaction with serum proteins. No detectable band for SA-CpG1668 was observed in gel analysis after SA-CpG1668/tripodna was mixed with plasma (Figure 4), indicating that the structure of SA-CpG1668/tripodna was maintained after the interaction with plasma components.

CpG-ODN-induced immune stimulation can be a function of the amount of CpG ODN interacting with TLR9 in endosomes after endocytic cellular uptake. The processes involved in this are interaction with cell membrane, endocytic uptake, dissociation into single-stranded CpG ODN (when needed), and interaction with TLR9. In general, lipid modification increases the cellular uptake of ODNs. Opti-MEM, a protein-free culture medium, was used for the in vitro experiments, using RAW264.7 cells (Figure 5 and Figure 6) in order to directly evaluate the effect of SA modification on the interaction of CpG1668/tripodna and SA-CpG1668/tripodna with RAW264.7 cells. In the present study, SA modification significantly increased the uptake of CpG1668 by RAW264.7 cells (Figure 5). Furthermore, the uptake amounts were higher when SA-CpG1668 was loaded into the tripodna. These results indicate that nanostructurization is a useful strategy for APC delivery not only for unmodified CpG ODNs, but also for lipid-conjugated CpG ODNs.

We also found that the Tm value of SA-CpG1668/tripodna was higher than that of CpG1688/tripodna (Table 2), thus indicating that SA-CpG1668/tripodna is more stable than CpG1688/tripodna. These results also indicate that SA-CpG1668 was less efficiently released from the nanostructure than CpG1668, which would reduce the concentration of free SA-CpG1668 in the endosome. Therefore, reducing the Tm values between SA-CpG1668 and its complementary ODN in tripodna would increase the concentration of free SA-CpG1668 in the endosome, leading to higher immune stimulation [35,36,37]. Further studies are needed to confirm this hypothesis.

The in vivo cytokine data showed that SA-CpG1668/tripodna increased the cytokine level selectively in the inguinal draining lymph nodes (Figure 7), with a decreasing trend in plasma (Figure 8). The difference in the IL-12p40 levels between SA-CpG1668 and SA-CpG1668/tripodna (Figure 7) strongly suggests that SA-CpG1668/tripodna maintained its structure even after in vivo administration. The IL-12p40 production in the lymph nodes after administration of SA-CpG1668/tripodna strongly suggests that SA-CpG1668 is released from SA-CpG1668/tripodna after uptake by the TLR9-positive cells in the lymph nodes, because TLR9 only recognizes single-stranded CpG ODNs. The results of the present study suggest that SA-CpG1668/tripodna binds to serum albumin and that the SA-CpG1668/tripodna and albumin complex efficiently reaches the lymph nodes, with reduced amount of SA-CpG1668 entering the systemic circulation (Figure 9). These distribution properties would be suitable for the application of CpG ODNs as immunostimulatory adjuvants, because increasing cytokine levels in the systemic circulation can lead to cytokine storm syndrome.

In the present study, tripodna, which is the simplest polypodna, was used to show the proof of concept of the combination of nanostructurization and lipid modification for efficient delivery of CpG ODN to draining lymph nodes. Because the delivery efficiency of polypodna to various types of cells, including APCs, highly depends on the number of pods, the more the pods of polypodna, the more efficient the cellular uptake. Therefore, replacing tripodna with more pods, such as hexapodna, would result in the development of a more efficient delivery system for CpG ODNs. Furthermore, a simultaneous delivery system of antigen and CpG ODN can be developed by combining an antigen and DNA hydrogel consisting of SA-conjugated CpG ODN.

## 4. Materials and Methods

### 4.1. Chemicals

Stearoyl chloride and 1-(3-dimethylaminopropyl)-3-ethylcarbodiimide hydrochloride (EDC-HCl) were purchased from Tokyo Chemical Industry Co., Ltd. (Tokyo, Japan). Triethylamine, tetrahydrofuran (THF) super dehydrated with stabilizer (super dehydrated), NHS, *N*, *N*-dimethylformamide (DMF), hydrogen chloride (HCl), and BSA were purchased from FUJIFILM Wako Pure Chemical Corporation (Osaka, Japan). Roswell Park Memorial Institute (RPMI) medium was obtained from Nissui Pharmaceutical Co., Ltd. (Tokyo, Japan), and fetal bovine serum (FBS) was obtained from Equitech-Bio, Inc. (Kerrville, TX, USA). Opti-modified Eagle’s medium (Opti-MEM) was purchased from Thermo Fisher Scientific K.K. (Tokyo, Japan). All other chemicals were of the highest grade available and used without further purification.

### 4.2. ODNs

Phosphodiester ODNs were purchased from Integrated DNA Technologies, Inc. (Coralville, IA, USA). CpG1668 with the same sequence as phosphorothioate CpG ODN 1668 was selected as CpG ODN, and 5′-NH_2_-CpG1668 was used for the synthesis of SA-CpG1668. The complementary sequence to the CpG motif (GACGTT) is AACGTC, but the sequence in Tri-1, Tri-2, and Tri-3 complementary to the motif was designed to AACTTC to avoid the presence of CG dinucleotide sequences in these ODNs.

### 4.3. Cells

Murine macrophage-like RAW264.7 cells were grown at 37 °C in the RPMI medium supplemented with 10% heat-inactivated FBS, 0.15% sodium bicarbonate, 100 units/mL penicillin, 100 mg/mL streptomycin, and 2 mM L-glutamine in humidified air containing 5% carbon dioxide (CO_2_). Cells were then seeded onto a 96-well culture plate at a density of 5 × 10^4^ cells/well and cultured for 24 h before use.

### 4.4. Animals

C57BL/6N mice (female, 6-week-old) were purchased from Sankyo Labo Service Co., Inc. (Tokyo, Japan). All animal experiments were conducted in accordance with the principles and procedures outlined in the National Institutes of Health Guide for the Care and Use of Laboratory Animals. The protocols for the animal experiments were approved by the Animal Experimentation Committee of the Tokyo University of Science. (Y21020; 30 April 2021).

### 4.5. Synthesis of SA-NHS

To an ice-cold solution of 63.3 mg (0.55 mmol; 1.1 eq.) *N*-hydroxysuccinimide and 55.7 mg (0.55 mmol; 1.1 eq), triethylamine in 5 mL dry THF, 151 mg (0.5 mmol; 1 eq) stearoyl chloride was added dropwise. The reaction mixture was stirred for 6 h at approximately 22 °C. After stirring, the solvent was removed via evaporation. The residue was dissolved in dichloromethane, washed with 30 mL of 0.1 M HCl, and then washed twice with 20 mL water. The organic phase was dried over sodium sulfate, and the solvent was removed to obtain SA-NHS. The structure was confirmed by ^1^H-NMR (Appendix A).

### 4.6. Synthesis of SA-CpG1668

After 5′-NH_2_-CpG1668 (10 nmol, 1 mM in 10 µL distilled H_2_O) was added to a mixture of NaH_2_PO_4_–NaHPO_4_ buffer (0.307 M, pH = 8.6, 18.5 µL) and DMF (40 µL), SA-NHS (20 µL, 55 mM in 20 µL DMF) was added to the mixture. After moderate mixing of all the reagents, using a vortex mixer and centrifugation, the mixture was stored overnight at 65 °C for further reaction. The reactant was purified by using a Zeba spin desalting column (7 K MWCO; Thermo Fisher Scientific Inc., Waltham, MA, USA), according to the manufacturer’s protocol. Thereafter, it was lyophilized to yield a white powder, which was then dissolved in distilled H_2_O. The formation of SA-CpG1668 was confirmed by MALDI–TOF-MS (JMS-S3000, JEOL Ltd.; MS Calcd = 6504.6; Found = 6504.0, Appendix A). Furthermore, the yield was confirmed by using a Nanodrop 2000/2000c (Thermo Fisher Scientific, Inc.).

### 4.7. Protein-Binding Assay

BSA was dissolved in 1× phosphate-buffered saline (PBS). Then 1 µL aliquot of 1 mM CpG1668 or SA-CpG1668 in 1× PBS was added to the desired number of equivalents of BSA in 1× PBS, and the final volume was made up to 10 µL with PBS. Binding to BSA was confirmed by PAGE (6%). ODNs were visualized by staining with ethidium bromide (EtBr; Nippon Gene Co., Ltd., Tokyo, Japan) and observed with an iBright FL1000 Imaging System (Thermo Fisher Scientific, Inc.). The plasma collected from the mice was serially diluted with 1× PBS. Diluted plasma was added to 1 µL aliquot of 1 µM SA-CpG1668/tripodna in 1× PBS, and the final volume was made up to 10 µL with PBS. Plasma-protein binding was confirmed by PAGE (6%).

### 4.8. Preparation of Tripodna

The tripodna was prepared as previously reported [23]. In brief, the three ODNs involved in the tripodna were mixed with or without CpG1668 or SA-CpG-1668 in appropriate molar ratios, heated to 95 °C, and then gently cooled to 4 °C.

### 4.9. PAGE

The formation of tripodna, CpG1668/tripodna, and SA-CpG1668/tripodna was confirmed by PAGE (6%), which was carried out at 200 V for 30 min at approximately 4 °C. A total of 100 ng of each DNA sample was added to the gel. The 20 bp DNA ladder was purchased from TaKaRa (Tokyo, Japan).

### 4.10. Measurement of Tm

Tm was measured by using a spectrophotometer (JASCO J-730; JASCO, Tokyo, Japan) in a quartz cell with an optical path length of 1 mm (JASCO). All samples were scanned from 20 to 95 °C at a rate of 1 °C/min, and the absorbance was recorded every 0.5 °C. Two types of buffer solutions were used for the measurement. One was a TE buffer solution (10 mM Tris, 1 mM ethylenediaminetetraacetic acid (EDTA), pH 7.4) containing 142 mM sodium, 4 mM potassium, and 5 mM calcium ions, and the other was a TE buffer solution (pH 5.5) containing 70 mM sodium and 32.5 mM potassium ions. The former was used as a simulated extracellular fluid, and the latter as a simulated endosomal fluid [38].

### 4.11. Cellular Uptake

Alexa Fluor568-labeled tri-1 was used to construct the Alexa Fluor568-labeled DNA samples. Cells were seeded into Lab-Tek chambered cover glass at a density of 2 × 10^4^ cells/well and incubated for 24 h before treatment. After aspiration of the supernatant, the DNA samples diluted with Opti-MEM were added to cells at 1.0 µM concentration of Alexa Fluor568-labeled tri-1. After 2 h of incubation at 5% CO_2_ and 37 °C, the cells were washed twice with PBS, fixed with 4% paraformaldehyde for 20 min, and washed again twice with PBS. Then 60 nM 4′,6-diamino-2-phenylindole (DAPI) was added to stain the nuclei, and the cells were incubated for 30 min at 4 °C. Then the cells were observed by using a confocal laser scanning microscope (Leica Microsystem, Mannheim, Germany).

### 4.12. Release of TNF-α from RAW264.7 Cells

Murine macrophage-like RAW264.7 cells were seeded onto a 96-well plate at a density of 5 × 10^4^ cells/well and incubated for 24 h before the experiment. DNA samples diluted in Opti-MEM were added to the cells at a concentration of 1 µM of CpG1668 or SA-CpG1668. After 8 h of incubation at 5% CO_2_ and 37 °C, the supernatant was collected and the concentration of TNF-α was determined by using enzyme-linked immunosorbent assay (ELISA) in accordance with the manufacturer’s protocol (BioLegend, Inc., San Diego, CA, USA).

### 4.13. IL-12p40 Production in Inguinal Lymph Nodes of Mice after Subcutaneous Injection

Mice were injected with CpG1668, SA-CpG1668, CpG1668/tripodna, or SA-CpG1668/tripodna in PBS at a dose of 10 nmol CpG/mouse into the subcutaneous tissue near the tail base. Eight hours after injection, the mice were euthanized by cervical dislocation, and the inguinal lymph nodes were collected. Lysis buffer was added to the lymph nodes, and the mixture was homogenized by using a homogenizer and then centrifuged at 12,000× *g* for 5 min. The concentration of the p40 subunit of IL-12 (IL-12p40) in the supernatant of the homogenate was determined by ELISA (BioLegend, Inc.), according to the manufacturer’s protocol.

### 4.14. IL-6 Production in Plasma of Mice after Subcutaneous Injection

Mice were injected with CpG1668, SA-CpG1668, CpG1668/tripodna, or SA-CpG1668/tripodna at a dose of 10 nmol CpG/mouse, as described above. At 1, 3, and 8 h after injection, blood samples were collected from the facial vein of the mice, and plasma was collected by centrifugation and stored at −20 °C until further analysis. The levels of IL-6 in the plasma were determined by ELISA, as described above.

### 4.15. Accumulation of PI-Labeled CpG1668/Tripodna and SA-CpG1668/Tripodna in Lymph Nodes after Subcutaneous Injection

CpG1668/tripodna and SA-CpG1668/tripodna were labeled with propidium iodide (PI). Mice were injected with PI-labeled CpG1668/tripodna or SA-CpG1668/tripodna in PBS at a dose of 10 nmol CpG/mouse into the subcutaneous tissue near the tail base. At 1, 3, and 6 h after injection, the mice were euthanized by cervical dislocation, and the axillary, inguinal, and popliteal lymph nodes were collected. The fluorescence intensity of the lymph nodes was detected by using an in vivo imaging system (In-Vivo Xtreme II, Bruker BioSpin, Billerica, MA, USA).

### 4.16. Statistical Analysis

Statistical differences were evaluated by one-way analysis of variance (ANOVA), followed by Tukey’s test for multiple comparisons or Student’s *t*-test for two groups. Statistical significance was set at *p* < 0.05.

## 5. Conclusions

We demonstrated that the combination of SA modification and nanostructurization is useful for delivering CpG ODNs to the draining inguinal lymph nodes and for increasing the cytokine production in the nodes with reduced levels of IL-6 in the systemic circulation. This suitable distribution was mediated by the interaction with plasma proteins, including serum albumin, through the SA molecule on CpG ODN. The findings of the present study will aid in the development of an efficient delivery system for CpG ODNs, with maximum benefits and minimum side effects, to be used as an immune adjuvant.

## Figures and Tables

**Figure 1 ijms-23-01350-f001:**
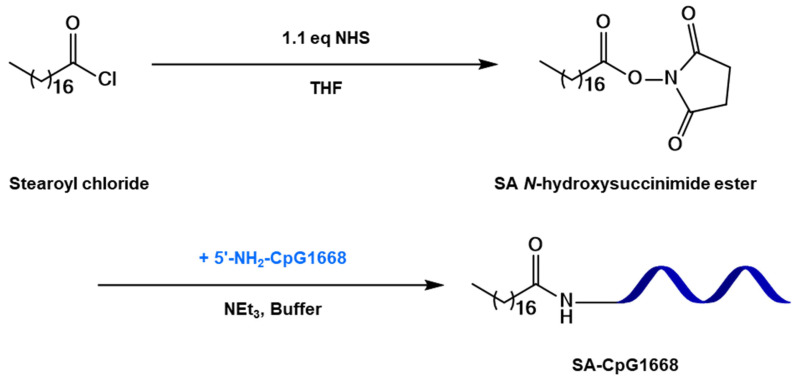
Synthesis scheme of SA-CpG1668. NHS, *N*-hydroxysuccinimide; THF, tetrahydrofuran; NEt_3_, triethylamine.

**Figure 2 ijms-23-01350-f002:**
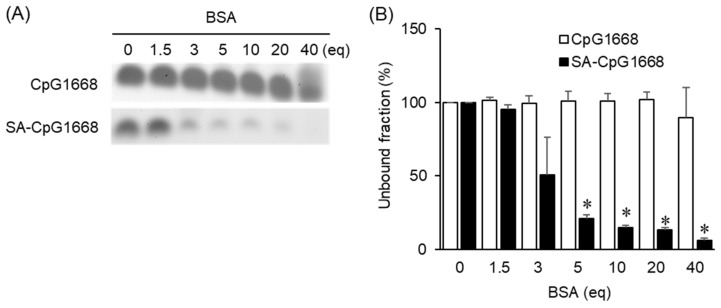
Polyacrylamide gel electrophoresis (PAGE) analysis of the binding of CpG1668 and SA-CpG1668 to bovine serum albumin (BSA). (**A**) CpG1668 or SA-CpG1668 (1 nmol/well) was mixed with 0–40 equivalents of BSA. The bands of free DNA are shown. Note: eq = equivalent. (**B**) The unbound fraction of DNA was calculated by using the iBright FL1000 Imaging System. * *p* < 0.05 vs. CpG1668 (*n* = 3).

**Figure 3 ijms-23-01350-f003:**
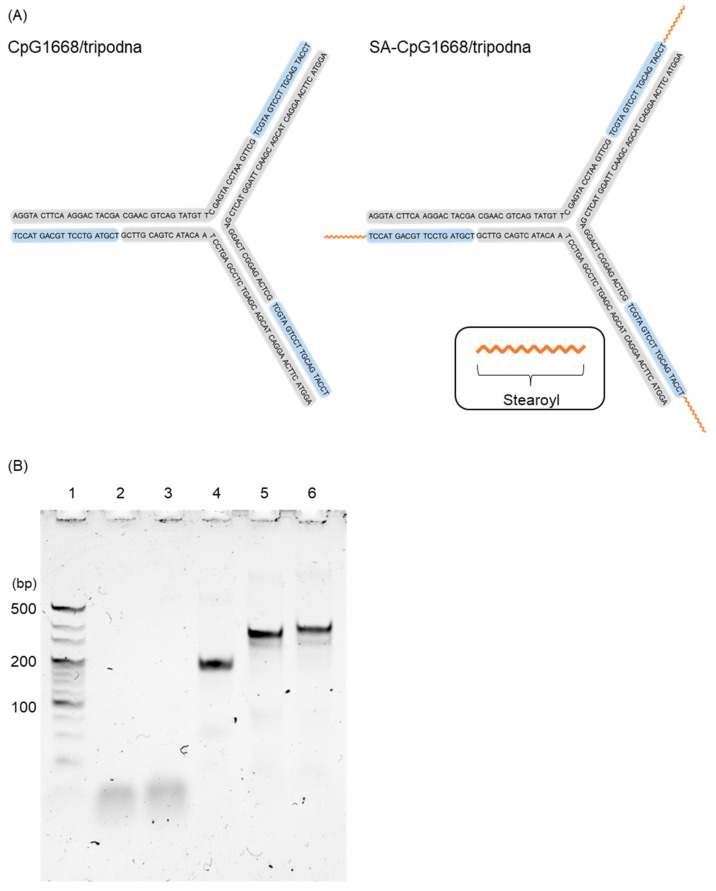
Construction and confirmation of CpG1668/tripodna and SA-CpG1668/tripodna. (**A**) Schematic representation of the putative structures of CpG1668/tripodna and SA-CpG1668/tripodna. (**B**) PAGE analysis of CpG1668, SA-CpG1668, CpG1668/tripodna, and SA-CpG1668/tripodna. DNA samples were run on a 6% polyacrylamide gel at 200 V for 30 min at 4 °C. Lane 1, 20 bp DNA ladder; lane 2, CpG1668; lane 3, SA-CpG1668; lane 4, tripodna; lane 5, CpG1668/tripodna; and lane 6, SA-CpG1668/tripodna.

**Figure 4 ijms-23-01350-f004:**
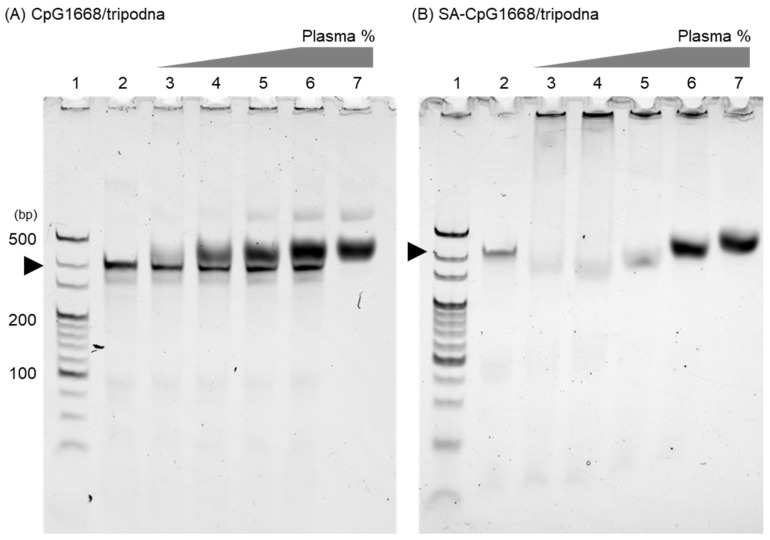
PAGE analysis of the binding of CpG1668/tripodna and SA-CpG1668/tripodna to plasma proteins. CpG1668/tripodna or SA-CpG1668/tripodna (0.1 pmol/well) was added to mouse plasma at varying ratios. Lane 1, 20 bp DNA ladder; lane 2, CpG1668/tripodna (**A**) or SA-CpG1668/tripodna (**B**) alone; lane 3, +20% plasma; lane 4, +40% plasma; lane 5, +60% plasma; lane 6, +80% plasma; and lane 7, 80% plasma only (no DNA).

**Figure 5 ijms-23-01350-f005:**
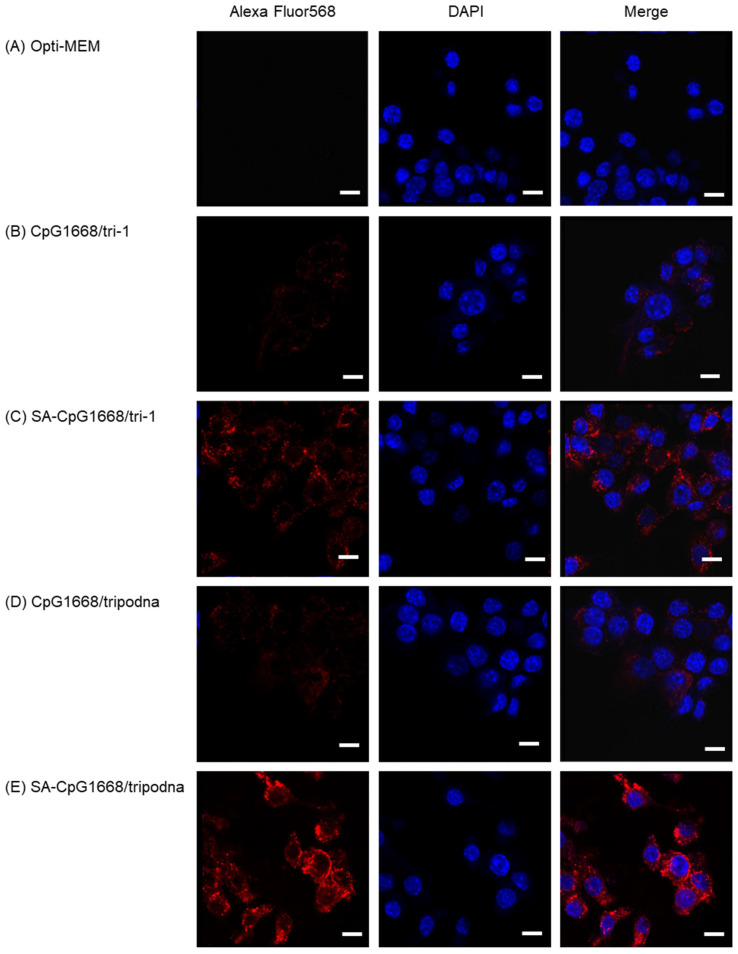
Uptake of DNA samples by RAW264.7 cells. Alexa Fluor568-labeled tri-1 was added to RAW264.7 cells at a final concentration 1.0 µM, and the cells were incubated for 2 h at 37 °C. (**A**) Opti-MEM, (**B**) CpG1668/Alexa Flour568-labeled tri-1, (**C**) SA-CpG1668/Alexa Flour568-labeled tri-1, (**D**) CpG1668/Alexa Flour568-labeled tripodna, and (**E**) SA-CpG1668/Alexa Flour568-labeled tripodna. Scale bars represent 10 µm.

**Figure 6 ijms-23-01350-f006:**
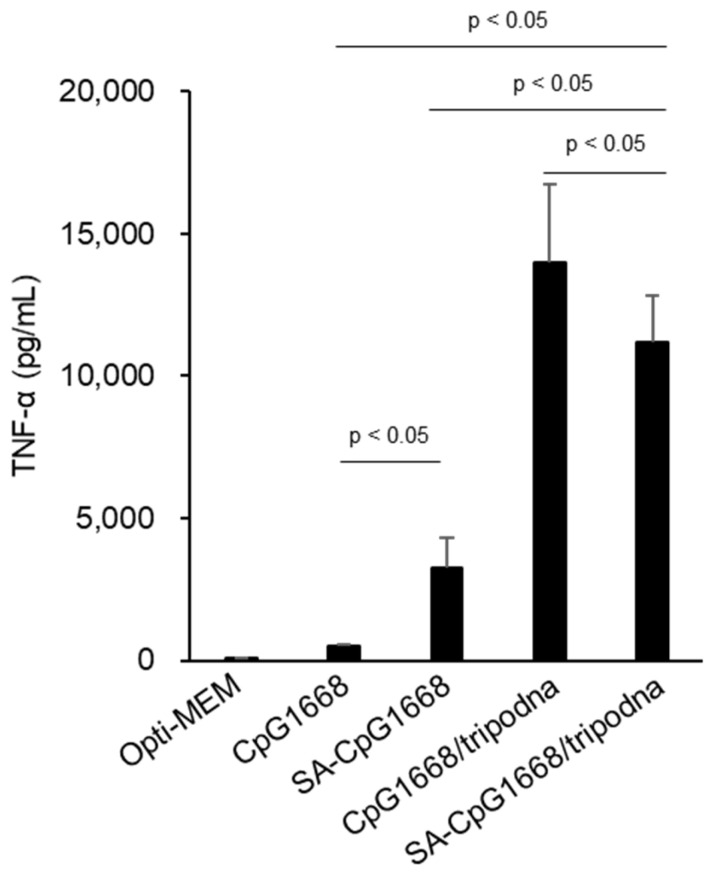
Release of tumor necrosis factor (TNF)-α from RAW264.7 cells. Each sample was added to RAW264.7 cells at a final concentration of 1.0 µM of CpG1668 or SA-CpG1668. After incubation for 8 h, the supernatant was collected, and the concentration of TNF-α was measured. Differences by Tukey’s test are shown. Results are expressed as the mean ± standard deviation (SD) (*n* = 4).

**Figure 7 ijms-23-01350-f007:**
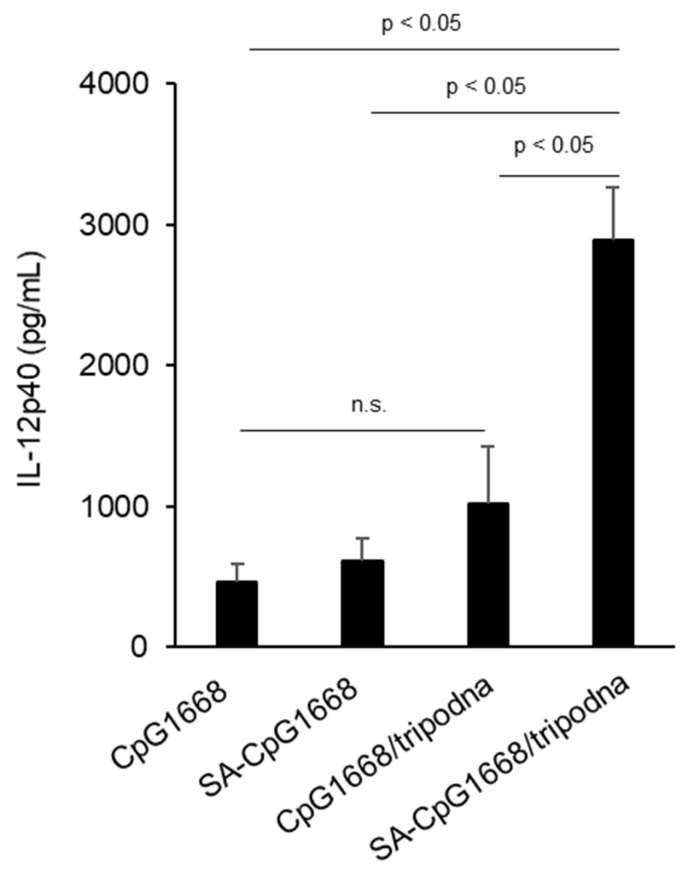
Interleukin (IL)-12p40 production in the draining inguinal lymph nodes after the subcutaneous injection of DNA samples at a dose of 10 nmol CpG/mouse. At 8 h after injection, mice were euthanatized by cervical dislocation and the inguinal lymph nodes were collected. The concentrations of IL-12p40 in lymph nodes were measured. Differences by Tukey’s test are shown. Results are expressed as the mean ± SD (*n* = 3); n.s., not significant.

**Figure 8 ijms-23-01350-f008:**
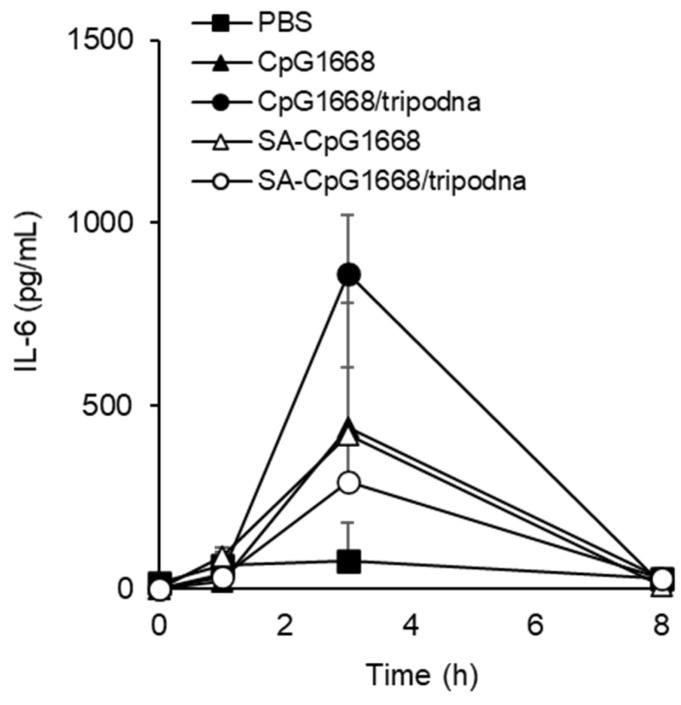
IL-6 concentration in mouse plasma after the subcutaneous injection of DNA samples at a dose of 10 nmol CpG/mouse. At 1, 3, and 8 h after injection, blood was collected, and the concentrations of IL-6 in plasma were measured. Results are expressed as the mean ± SD (*n* = 3).

**Figure 9 ijms-23-01350-f009:**
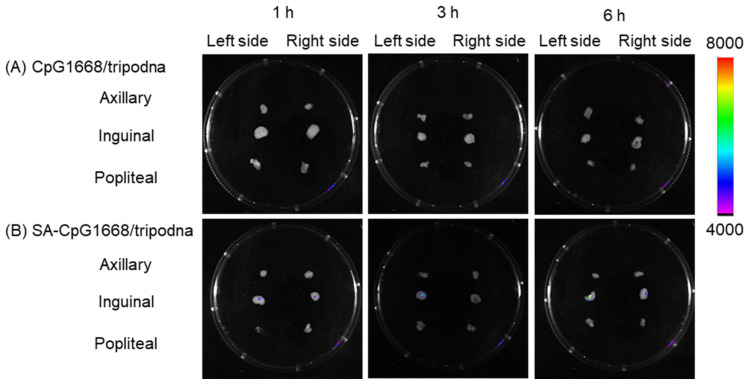
Accumulation of PI-labeled CpG1668/tripodna and SA-CpG1668/tripodna in lymph nodes after subcutaneous injection to mice. At 1, 3, and 6 h after injection, the axillary, inguinal, and popliteal lymph nodes were collected and observed by in vivo imaging system. (**A**) PI-labeled CpG1668/tripodna and (**B**) SA-CpG1668/tripodna.

**Table 1 ijms-23-01350-t001:** Sequences of ODNs.

ODN	Sequence (5′ to 3′)
CpG1668	TCCATGACGTTCCTGATGCT
SA-CpG1668	Stearoyl-TCCATGACGTTCCTGATGCT
Tri-1	GCTTGAATCCATGAGCTTGTATGACTGCAAGCAGCATCAGGAACTTCATGGA
Tri-2	GCTTGCAGTCATACAATCCTGAGCCTCTGAGCAGCATCAGGAACTTCATGGA
Tri-3	GCTCAGAGGCTCAGGAGCTCATGGATTCAAGCAGCATCAGGAACTTCATGGA

All ODNs have a phosphodiester backbone. The underlined complementary sequence of the CpG motif (GACGTT) was designed to have a single nucleotide mismatch (AACTTC) to avoid TLR9 stimulation by these ODNs.

**Table 2 ijms-23-01350-t002:** Tm of the DNA Samples.

Sample	Tm (°C)	
	pH 7.4	pH 5.5
CpG1668/tri-1	52.2	49.8
SA-CpG1668/tri-1	61.0	58.8
Tripodna	65.6	59.3
CpG1668/tripodna	59.8	56.0
SA-CpG1668/tripodna	63.5	60.2

Buffer solutions with the following composition were used: pH 7.4, a TE buffer (10 mM Tris, 1 mM EDTA) containing 142 mM sodium, 4 mM potassium, and 5 mM calcium ions; pH 5.5, a TE buffer (10 mM Tris, 1 mM EDTA) containing 70 mM sodium ion and 32.5 mM potassium ion.

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
