# Peer review of "Targeted Delivery of Immunostimulatory CpG Oligodeoxynucleotides to Antigen-Presenting Cells in Draining Lymph Nodes by Stearic Acid Modification and Nanostructurization"

_ijms, 2022, doi:10.3390/ijms23031350_

Round 1

Reviewer 1 Report

The paper reports on the effect of stearic acid conjugation to 5'-NH2 of CpGODNs, the complex of cytosine-phosphate guanine-oligodeoxynucleotides, which make complex with polyposid-like structured nucleic acids (poly-podna) on the effective binding to albumin and the effective delivery to antigen-presenting cells (APCs). The results are interesting, but the experiment in the title has not been done and there seems to be problems with the design of the experiment.

  1. The authors should describe the structural information for CpG 1668/tripodna and SA-CpG 1668/tripodna and indicate that the SA is facing out.
  2. In Figure 4 (A), smeared broad bands are observed on Lanes 4, 5, 6 and 7 of the CpG 1668/tripodna and (B) on Lanes 6 and 7. What are they?
  3. In in vitro experiments, DNA samples were diluted with Opti-MEM, but explain how CpG 1668/tripodna and SA-CpG 1668/tripodna exist in Opti-MEM? Why didn’t the authors use albumin or serum protein solutions?
  4. In relation to 3, CpG 1668/tripodna and SA-CpG 1668/tripodna were examined as being present in different states in vitro and in vivo. The reviewer worries about this.
  5. It is essential to show that CpG 1668/tripodna and SA-CpG 1668/tripodna are actually delivered to lymph nodes in vivo.
  6. Line 212 says “Previous studies on ODN modified with cholesterol or fatty acids have shown that lipid-modified ODNs form micelles in aqueous media.” Line 213 says “SA-CpG 1668 hardly formed micelles in PBS”. Is SA fatty acid?
  7. If the authors would like to discuss about the state of presence, they should look into whether SA-CpG 1668/tripodna is bound to albumin or not, and SA-CpG 1668 is released from SA-CpG 1668/tripodna to interact with APCs or not. Namely, the dynamics of SA-CpG 1688 should be the target of this study.

Reviewer 2 Report

Dear authors,

I have read the article, Targeted Delivery of Immunostimulatory CpG Oligodeoxynu- 2 cleotides to Antigen-Presenting Cells in Draining Lymph 3
Nodes by Stearic Acid Modification and Nanostructurization' with high interest. The article is novel, experiments are well carried out and presented well. Hence I am recommending to accept this manuscript after major revisions:

  1. No statistical analysis were performed for Fig.2B and Fig 8.
  2. Evidence of nanostructurization is required either by AFM, TEM.
  3. What is the size of these nanocomplex? and what is the surface charge or zeta potential?
  4. Please provide NMR, Mass/Maldi data for SA-NHS, SA-CpG1668
  5. Authors needs to show nanocomplex stability in various buffers including 10% plasma, phosphate buffer solution. 
  6. Please refer: Reddy et al, Nanoscale (2019), 11, 7931-7943

Round 2

Reviewer 2 Report

The authors have revised the manuscript as per reviewers comments and the present version can be accepted for publication in IJMS.